# Profound Properties of Protein-Rich, Platelet-Rich Plasma Matrices as Novel, Multi-Purpose Biological Platforms in Tissue Repair, Regeneration, and Wound Healing

**DOI:** 10.3390/ijms25147914

**Published:** 2024-07-19

**Authors:** Peter A. Everts, José Fábio Lana, Robert W. Alexander, Ignacio Dallo, Elizaveta Kon, Mary A. Ambach, André van Zundert, Luga Podesta

**Affiliations:** 1Gulf Coast Biologics, A Non-Profit Organization, Fort Myers, FL 33916, USA; 2OrthoRegen Group, Max-Planck University, Indaiatuba 13334-170, SP, Brazil; josefabiolana@gmail.com; 3Regenevita Biocellular Aesthetic & Reconstructive Surgery, Cranio-Maxillofacial Surgery, Regenerative and Wound Healing, Hamilton, MT 59840, USA; rwamd1914@gmail.com; 4Department of Surgery & Maxillofacial Surgery, School of Medicine & Dentistry, University of Washington, Seattle, WA 98195, USA; 5Unit of Biological Therapies and MSK Interventionism, Department of Orthopaedic Surgery and Sports Medicine, Sport Me Medical Center, 41013 Seville, Spain; doctorignaciodallo@gmail.com; 6Department of Biomedical Sciences, Humanitas University, Pieve Emanuele, 20072 Milan, Italy; elizaveta.kon@humanitas.it; 7IRCCS Humanitas Research Hospital, Rozzano, 20089 Milan, Italy; 8BioEvolve, San Diego Orthobiologics and Sports Center, San Diego, CA 92024, USA; 9Department of Anaesthesia and Perioperative Medicine, Royal Brisbane and Women’s Hospital, Brisbane and The University of Queensland, Brisbane 4072, Australia; a.vanzundert@uq.edu; 10Bluetail Medical Group & Podesta Orthopedic Sports Medicine, Naples, FL 34109, USA; lugamd@aol.com; 11Physical Medicine & Rehabilitation Orlando College of Osteopathic Medicine, Orlando, FL 32806, USA

**Keywords:** protein-rich platelet concentrate, matrix, platelet-rich plasma, ultrafiltration, tissue repair, tissue regeneration, fibrin(ogen)

## Abstract

Autologous platelet-rich plasma (PRP) preparations are prepared at the point of care. Centrifugation cellular density separation sequesters a fresh unit of blood into three main fractions: a platelet-poor plasma (PPP) fraction, a stratum rich in platelets (platelet concentrate), and variable leukocyte bioformulation and erythrocyte fractions. The employment of autologous platelet concentrates facilitates the biological potential to accelerate and support numerous cellular activities that can lead to tissue repair, tissue regeneration, wound healing, and, ultimately, functional and structural repair. Normally, after PRP preparation, the PPP fraction is discarded. One of the less well-known but equally important features of PPP is that particular growth factors (GFs) are not abundantly present in PRP, as they reside outside of the platelet alpha granules. Precisely, insulin-like growth factor-1 (IGF-1) and hepatocyte growth factor (HGF) are mainly present in the PPP fraction. In addition to their roles as angiogenesis activators, these plasma-based GFs are also known to inhibit inflammation and fibrosis, and they promote keratinocyte migration and support tissue repair and wound healing. Additionally, PPP is known for the presence of exosomes and other macrovesicles, exerting cell–cell communication and cell signaling. Newly developed ultrafiltration technologies incorporate PPP processing methods by eliminating, in a fast and efficient manner, plasma water, cytokines, molecules, and plasma proteins with a molecular mass (weight) less than the pore size of the fibers. Consequently, a viable and viscous protein concentrate of functional total proteins, like fibrinogen, albumin, and alpha-2-macroglobulin is created. Consolidating a small volume of high platelet concentrate with a small volume of highly concentrated protein-rich PPP creates a protein-rich, platelet-rich plasma (PR-PRP) biological preparation. After the activation of proteins, mainly fibrinogen, the PR-PRP matrix retains and facilitates interactions between invading resident cells, like macrophages, fibroblast, and mesenchymal stem cells (MSCs), as well as the embedded concentrated PRP cells and molecules. The administered PR-PRP biologic will ultimately undergo fibrinolysis, leading to a sustained release of concentrated cells and molecules that have been retained in the PR-PRP matrix until the matrix is dissolved. We will discuss the unique biological and tissue reparative and regenerative properties of the PR-PRP matrix.

## 1. Introduction

Over the last several decades, biologically derived materials and tissues have evolved in many different forms, like coatings, scaffolds, biosensors, and functional biomaterials [1,2]. Initially, it was thought to substitute damaged human body parts. Later, the focus shifted from the use of materials to biology, giving rise to the development of tissue engineering concepts [3,4,5].

Regenerative medicine technology and clinical applications emerged from several clinical practices like bone grafting, surgical implants, scaffold-based biomaterials, and bone marrow transplantations [1]. The use of autologous biological preparations, including PRP and fibrin-based scaffolds, has gained remarkable momentum. A wide range of medical applications use autologous platelet-rich plasma (PRP) formulations and plasma protein-based biological preparations for non-surgical tissue repair, regeneration, and wound healing applications [6,7]. The employment of PRP and protein-based technologies has generated promising patient outcomes, as these complex biological preparations and processes involve a multitude of local and systemic cellular and molecular activities in a sequential manner aimed at restoring the integrity and function of damaged tissues.

PRP preparations employ centrifuges and gravitational cellular density separation protocols to separate a unit of fresh whole blood into a platelet-poor plasma (PPP) fraction, a platelet-buffy coat stratum, with or without leukocytes, and a red blood cell (RBC) fraction [8].

The term “platelet-rich plasma” was first used in 1954 by Kingsley et al. and referred to the standardization of platelet concentrate preparations for transfusion [9]. In 1972, for the first time, Matras used platelets as sealants to improve tissue healing following surgical procedures [10]. Thereafter, an autologous product termed “platelet–fibrinogen–thrombin mixture” was developed, including, in fibrin glue, a significant concentration of platelets to reinforce fibrin polymerization in corneal wounds [11]. In the following years, Knighton et al. described the role of platelets in wound-healing procedures by using a blood-derived product called “platelet-derived wound healing factors” [12]. The term platelet gel was introduced by Whitman and associates [13] in oral and maxillofacial surgery. In contrast, the term PRP in regenerative medicine was introduced by Marx et al. in 1998, identifying the potential for tissue healing using platelet growth factors [14]. Thereafter, the term PRP was generically associated with many different formulations of platelet concentrates, including the denser fibrin-based PRP [15].

The PRP-induced tissue repair, regeneration, and wound-healing potential are based on the release of a plethora of platelet constituents from alpha, dense, and lysosomal granules, as well as the activities of many platelet adhesion molecules. Furthermore, the absence, or presence, of particular leukocytes has an effect on platelet cellular functions, like angiogenesis, immunomodulation, cell signaling and metabolism, and ultimately, tissue repair and functional tissue restoration [16,17].

Regretfully, there is no consensus on a comprehensive PRP-classification system or on autologous biological preparations, despite the latest attempt of Kon et al. presenting an expert opinion of a novel PRP classification system [18], resulting in an oversaturation of PRP devices with numerous preparation methods. Therefore, PRP functional characteristics, such as centrifugation protocols, platelet dosing, leukocyte and RBC content, and cellular delivery methods to treat specific pathological tissue conditions, differ substantially [19,20].

There are three main variables in a simplified classification system, namely, platelets, leukocytes, and plasma proteins [15]. In terms of these biological variables, PRP can be classified into four general categories [21]. A number of variables differentiate PRP devices, including the volume of processed whole blood; the device platelet capture rate, which influences the platelet concentration; and the prepared PRP volume. In addition, the presence and concentration of leukocytes is an underestimated factor in PRP treatment outcomes. Accordingly, the literature denotes poor patient outcomes when PRP devices with low platelet numbers and inconsistent other cellular content were used [22,23,24,25].

Following PRP preparation, the platelet concentrate is removed from the device for clinical application, while the PPP fraction, with its own molecular and cellular components, is usually discarded as it is not considered therapeutically beneficial. In spite of this, certain growth factors, such as insulin-like growth factor-1 (IGF-1) and hepatocyte growth factor HGF), are predominantly found in the PPP fraction outside of platelets [26,27]. The PPP fractions also contain exosomes and other macrovesicular structures that are essential for cell-to-cell communication and cell signaling [28,29].

The advancement of technology has led to the development of new concepts for biological preparations, such as the concentration of PPP via modified ultrafiltration in order to increase the plasma levels of plasma proteins, such as fibrinogen, alpha 2 macroglobulin (A2M), IGF-1, HGF, cytokines, and other biomolecules [27,29]. Modern low-volume ultrafiltration devices, with a small membrane pore size, create a protein-rich product that can be used as a standalone product or as a combination with high-definition PRP formulations to form a consolidated multicellular protein-rich platelet concentrate (PR-PRP) preparation. In comparison to PRP alone, PR-PRP contains a greater amount of diverse growth factors and other important non-platelet derivates [27], with several studies reporting synergistic effects between platelet-derived growth factors and non-platelet-derived growth factors [30,31]. PR-PRPs function as a biological matrix once activated, retaining and facilitating interactions between the matrix cellular and molecular constituents [29]. Moreover, the versatility of PR-PRP technology facilitates the creation of patient-specific formulations for non-surgical as well as surgical applications.

Our objective in this review is to provide a comprehensive overview of a multipurpose biological platform comprising high-definition PRP and viable protein-rich plasma, as well as their typical biological contributions to tissue repair, regeneration, and wound healing.

## 2. The Impact of PRP in PR-PRP

In an era in which traumatic and degenerative diseases are becoming more prevalent, regenerative medicine therapies aim to develop strategies to repair and regenerate damaged tissues to restore normal function [32].

Regenerative medicine strategies are primarily focused on non-surgical and interventional approaches, treating conditions where conventional therapies are ineffective or insufficient, using autologous biological preparations (ABP) that are capable of tissue repair and regeneration. In biosurgical procedures, regenerative medicine technologies are used to support post-surgical wound healing to minimize complications and improve clinical outcomes through the release of many biologically active components present in numerous autologous preparations, like several PRP formulations, bone marrow concentrates, and a variety of adipose tissue preparations [33,34]. In this review, we focus merely on PRP bioformulations and their rationale in PR-PRP.

### 2.1. Rationale for PRP Therapies

Physiological pathways can be stimulated following the precise delivery of concentrated platelets into degenerated and injured tissues to initiate tissue repair through the release of many platelet-derived growth factors, cytokines, molecules, and adhesive proteins. Additionally, local tissue regulators act through cell signaling, paracrine, autocrine, and intracrine mechanisms after PRP administration to facilitate tissue repair.

### 2.2. PRP Classification

PRP is a centrifugated and processed liquid fraction of harvested fresh peripheral blood obtained by gravitational density separation and is characterized by a heterogeneous and complex composition of multi-cellular components and a significant increase in platelet concentration compared to baseline values (Figure 1). Currently, the recommended dose of platelets in PRP is >1 billion platelets per mL [35,36].

Due to the lack of consensus regarding standardization and preparation methods, PRP devices may be categorized as pure PRP (P-PRP), leukocyte-poor PRP (LP-PRP), leukocyte-rich PRP (LR-PRP), and varying compositions of plasma-based preparations, such as platelet-rich fibrin (PRF) [21]. However, PRP and plasma-based PRP preparations differ significantly in composition, as demonstrated by a proteomic study [37]. Encouraging results following PRP applications have been reported [38]. Nevertheless, poor and even negative patient outcomes have also been reported, with PRP platelet dosing and cellular characterization being the most prevalent variables involved, largely due to differences in PRP device architecture and preparation techniques [39].

### 2.3. Platelet Structures and Biological Content

Platelets are anucleate and small cells pinched off from megakaryocytes in the red bone marrow before they are released on a continuous basis into the peripheral circulation on a continuous basis [40]. The platelet cell membrane is made of phospholipids and glycoproteins that interact with various ligands, like collagen and thrombin, and mediate platelet adhesion and aggregation [41]. Furthermore, glycoproteins facilitate cellular signaling to interact with extracellular molecules and cells [42].

Platelets contain three major types of secretory granules, including α-granules, dense granules, and lysosomes, containing unique cellular and molecular composites [43]. Proteomic-based analysis provided detailed information on the platelet granular content following platelet activation [37]. Dense granules and α-granules are the most well-studied and the most physiologically important, as they are involved in numerous cellular activities related to tissue repair and regeneration [35]. An overview of platelet structures, their main content, and other platelet elements is shown in Table 1.

#### 2.3.1. α-Granules

α-granules are unique to platelets, and they are they are most abundantly present, with approximately 50–80 granules per platelet, accounting for 10% of its volume [44].

Platelet proteomic studies have acknowledged that there are more than 300 soluble proteins in platelets [45]. Platelet α-granules contain a variety of proteins, including numerous platelet growth factors (PGFs) with their typical isoforms, like vascular endothelium growth factor (VEGF), platelet-derived growth factor (PDGF) with their various isoforms, transforming growth factor-beta (TGF-β), fibroblast growth factor (FGF), epidermal growth factor (EGF), and, to a lesser extent, hepatocyte growth factor (HGF) and insulin-like growth factor-1 (IGF-1) [6]. Furthermore, a variety of angiogenetic protein regulators, coagulation factors, and immunomodulatory molecules are stored in the alpha granules, as well as many chemokines and cytokines [45]. They serve as pro-inflammatory and immune-modulating factors when they are secreted, among others, regulated by T-Cell activation and probably secreted by T-Cells (RANTES) and platelet factor 4 (PF-4), also known as chemokine ligand 4 (CXCL4) [45,46]. Additionally, α-granules contain numerous platelet membrane-bound protein receptors, like P-selectin and integrins, that are expressed on the platelet surface [47].

##### Platelet-Derived Exosomes

Activated PRP will lead to the secretion of platelet granules as well as the release of extracellular vesicles (EVs), including exosomes, originating from intracytoplasmic multivesicular bodies (MVBs) [48,49]. In recent years, exosomes have received a lot of attention in regenerative medicine applications as active biological carriers in tissue repair and regeneration. Exosomes are secreted by many different metabolically active cells, like immune cells and mesenchymal stem cells (MSCs), and they are found in blood, bone marrow, synovial fluid, and urine [50,51,52,53]. Exosomes are small, differently sized EVs, 30–150 nm in diameter, and act as carriers of, among others, many biologically active proteins, mRNA, miRNA, and other bioactive substances [54,55].

Exosomes have a significant impact on many cellular functions, including the immune response and cell signaling [56,57], thereby playing crucial roles in intercellular communication [58] and platelet–cell communication [59], transporting their cargo between cells and organs during normal biological processes.

Platelet-derived exosomes (PLT-Exos) are stored in platelet alpha granules and MVBs [60]. The concentration of exosomes in plasma has been reported to range from 0.9 to 1.3 × 10^9^/mL [61], with PLT-Exos constituting approximately 75% of total plasma exosomes [62].

Torreggiani et al. isolated exosomes from PRP-platelets (PRP-Exos) and demonstrated their potential effects on the proliferation and differentiation of bone marrow MSCs, advocating their roles in tissue regeneration. It has been suggested that PRP-Exos may deliver a nano-delivery system, contributing to the healing mechanisms of PRP [63]. Interestingly, several released PGF can be encapsulated into exosomes in high concentrations and transported across the extracellular space to target tissues capable of promoting angiogenesis and fibroblast proliferation [64,65,66].

The potential for exosomes and other EVs in tissue repair, regeneration, and wound healing is extensive. However, a better understanding and optimization of PRP-Exos preparation methods and their function in PGF transportation need to be further explored and positioned against commercial purification processes.

#### 2.3.2. Dense Granules

The second most prevalent intra-platelet structure is dense granules. These storage organelles contain small molecules such as histamine, ADP, polyphosphates, serotonin (5-hydroxytryptamine, 5-HT), and epinephrine [67]. Their key function is converting platelets into an active state, initiating thrombus formation with the subsequent release of α-granular constituents.

It is also noteworthy that several secretory molecules are capable of immunomodulation [68]. Particularly, platelet ADP is connected to dendritic cellular activity, potentially resulting in increased antigen endocytosis and steering an immune response by linking the innate and adaptive immune systems [69]. Importantly, 5-HT is stored at high concentrations in dense granules and has been recognized as an important neurotransmitter involved in neuropsychological processes. Furthermore, 5-HT contributes to the regulation of several biological functions, including pain, inflammation, and immunomodulation [70,71,72].

#### 2.3.3. Lysosomes

Platelet lysosomal functions have not been well studied. Few studies mention the in vivo release of platelet lysosomal content, which contains an array of acid glycohydrolases, as reflected in Table 1. These enzymes can destroy glycoproteins and glycosaminoglycans [73], and they act as contributors to the cell’s digestive system, destroying materials taken up from outside the cell and digesting archaic cytosolic components [74]. Furthermore, Heijnen and van der Sluis mentioned lysosomal activities in extracellular functions, such as fibrinolysis, supportive in vasculature remodeling, and the degradation of extracellular matrix components [75].

On this matter, lysosome proteases have been implicated in tendon-homeostasis mechanisms due to their ability to cleave the tendon ECM [76]. Interestingly, since lysosomes contain proteases and cationic proteins with bactericidal activities, they interact as well with macrophages in phagocytosis [77].

### 2.4. Variability in Leukocyte Presence in PRP and PR-PRP

Throughout the years, practitioners and scientists have defined various formulations of PRP, including P-PRP and LP-PRP preparations, which result in minimal leukocyte contamination. Others refer to PRP as a biological preparation containing increased levels of platelets, leukocytes, or LR-PRP [6,15,78]. Furthermore, another classification of PRP is platelet-rich fibrin (PRF) or platelet–leukocyte-rich fibrin (L-PRF), containing greater levels of fibrin and bioactive proteins [15]. Specifically, with regard to the various PRP formulations, neutrophils, lymphocytes, monocytes, and macrophages have a significant role in the intrinsic biology of classical wound-healing cascades and have an impact on chronic tissue pathologies as a result of their ability to interact with platelets, immune cells, and host defense mechanisms [6,79].

Currently available PRP devices and technologies produce a broad range of bioformulations with regard to the platelet dosage, leukocyte populations, and concentrations, potentially resulting in varying effects on inflammation, immunomodulation, angiogenesis, nociception, and ultimately, tissue repair, regeneration, and wound healing [39,80,81].

The role of leukocytes in PRP preparations, as well as their impact on tissue regeneration and repair, is still widely underestimated. Preferably, PRP bioformulations should be substantiated based on the cellular properties of pathological tissue structures.

### 2.5. Tissue Repair and Regeneration Provoked by PRP

Tissue regeneration and repair utilizing PRP technology following degenerative processes or trauma is a complex biological process that involves numerous biological components and mechanisms. One of the key aspects in this process is the role of platelets secreting various substances to incite immunomodulation and angiogenesis and modulate pain to promote tissue repair and restore tissue functionality.

#### 2.5.1. Anti-Inflammatory Effects

The activation of platelets (e.g., by P-selectin) during acute and chronic inflammatory tissue conditions induces the release of platelet growth factors, pro- and anti-inflammatory cytokines, chemokines, and other biomolecules which act as chemotactic agents for neutrophils and monocytes [6].

A study conducted by Djenek et al. revealed that platelets and leukocytes present in LR-PRP interact with each other to decrease inflammation, inhibiting the expression of major inflammatory cells such as IL-1, IL-6, and TNF-α [82]. Furthermore, in the presence of LR-PRP, monocytes migrate to tissue inflammatory sites, where they polarize toward the anti-inflammatory macrophage phenotype 2, stimulating the release of IL-4 and IL-10 [83,84], and modify chemotaxis and proteolytic processes by reducing TNF-α production [85]. Additionally, platelet activation induces regulatory T-Cell activation and the release of specific platelet-derived chemokines, such as PF-4, RANTES, IL-1, and CXCL-12, to prevent monocytes from undergoing spontaneous apoptosis and promotes their differentiation into macrophages [86]. The release of platelet releasates also inhibits nuclear factor kappa B (NF-κB) activity, therefore reducing pro-inflammatory responses, as seen in synovial cells [87].

#### 2.5.2. Immunomodulation

The innate immune system identifies imposing microbes and fragments from tissue damage fragments when surface-expressed pattern-recognition receptors, like Toll-like receptors (TLRs) [88], bind to damaged and pathogen-associated molecular patterns. Because of TLR activation, NF-κB is activated, which regulates innate and adaptive immune responses. An adaptive immune system’s function follows the identification of pathogens or tissue damage by employing antigen-specific receptors to eliminate the pathogens.

Platelets are essential cells of both the innate and adaptive immune systems. They are among the first cells to identify microbial pathogens and are proficient in identifying endothelial injury [89]. In compromised tissues, non-activated platelets become rapidly activated and undergo exocytosis, releasing their granular and molecular content and expressing platelet chemokine receptors, immediately followed by platelet aggregation and the subsequent initiation of inflammatory pathways [90].

When PRP, consisting of a high concentration of platelets, is delivered to pathological tissues, a similar cascade of events is elicited with platelets interacting with local immune cells that express several surface immunomodulatory receptor molecules, such as platelet-specific TLRs, P-selectin, and other cytokines to regulate and optimize tissue repair, wound healing, and halt degeneration [91,92]. As opposed to LP-PRP formulations, LR-PRP contains leukocytic cells, which may initiate early phase immune responses in response to platelet activation [93]. Specifically, TLR-4 induces various interactions between platelets and neutrophils [94], including neutrophil degranulation with the release of reactive oxygen species and the creation of neutrophil-extracellular traps (NETs) that are capable of trapping and destroying pathogens through the process of NETosis [95]

Similarly, as a result of platelet activation, monocyte and macrophage effector functions are modulated, contributing to tissue inflammation and the differentiation of immune cells, and they initiate T- and B-cell responses via dendritic cells [96,97]. Furthermore, monocytes migrate into degenerative and diseased tissues and are able to alter chemotaxis and modify proteolysis by adhering to and secreting inflammatory mediators [98].

#### 2.5.3. Angiogenesis

During wound healing, tissue repair, and regeneration, optimization of angiogenetic pathways is required [39]. Angiogenesis in pathological microenvironments involves the sprouting and organization of micro vessels from pre-existing blood vessels [99]. It requires the restoration of numerous angiogenic mechanisms to overcome low oxygen tensions, low pH levels, and high lactate levels [39].

As a result of PRP therapy, PGFs and platelet pro- and anti-angiogenic factors mediate cell–cell communication and cell–matrix interactions, contributing to the formation of functional blood vessels [100,101]. PRP formulations containing a high platelet dose (>1.5 × 10^6^ platelets/µL), and therefore containing a high concentration of proangiogenic VEGF, were found to be most effective in stimulating angiogenesis [102]. Moreover, it has been demonstrated that the synergy between PDGF-BB and VEGF enhances the formation of a mature vascular network [103].

Activated leukocytes in LR-PRP are vigorously involved in angiogenesis [104]. Neutrophils, monocytes, and lymphocytes produce MMP-2 and MMP-9, contributing to the proliferation and migration of endothelial cells (ECs) [104]. Interestingly, matrix metalloproteinases (MMPs) are also released by platelets on a platelet dose-dependent basis, increasing the total concentration of MMPs [103]. Additionally, monocyte-derived macrophages secrete significant amounts of VEGF-A, which is important in tissue repair-induced angiogenesis [105]

According to Yuan et al., LR-PRP generates greater angiogenesis than LP-PRP, and the blood supply is more abundant when LR-PRP is applied [106]. The restoration of blood flow is essential during tissue-repair processes as new blood vessels can deliver oxygen and nutrients while simultaneously removing catabolic byproducts [107].

#### 2.5.4. Analgesic Effects

Platelet dosing and the PRP bioformulations have been identified as important features in PRP preparations to contribute to analgesic effects [108,109]. Obviously, patients who experience less pain tend to have better outcomes and more effective post-operative rehabilitation results.

Different mechanisms may be involved in the induction of painkilling mechanisms by PRP, both direct and indirect. According to Mohammadi et al., the physiological aspects of wound pain in post-surgical wound care patients were related to vascular injury and skin tissue hypoxia [107]. However, the application of PRP facilitated the reinstitution of neo-angiogenesis, resulting in improved tissue oxygenation and nutrient delivery, and patients experienced the benefit of reduced pain as a result.

Similar painkilling observations were reported in a systematic review regarding PRP utilization in orthopedic surgical and non-surgical spine procedures [109,110]. In another study, higher platelet concentrations were significantly associated with greater analgesic responses and higher post-treatment outcome satisfaction [111]. The impact of the platelet concentration on painkilling was also demonstrated in a small animal study. PRP containing a platelet concentration of 1 × 10^6^/µL completely relieved pain, while a reduction in platelet concentration by 50% induced significantly less pain relief [112].

In 2008, a mechanistic role for platelet-derived 5-HT was implied by Everts et al. [34]. They hypothesized that 5-HT, a critical neurotransmitter, plays a variety of well-defined analgesic roles in the central nervous system, with specific functions for the many 5-HT receptors [113]. Further, they hypothesized that administered PRP, containing huge amounts of 5-HT, stored in platelet-dense granules, may contribute to analgesic effects, as Sprott et al. reported significant improvements in pain scores associated with a significant reduction in platelet 5-HT levels and a concomitant increase in serum serotonin levels [114]. It is noteworthy that Kuffler found that several PRP-preparation aspects had varying effects on chronic pain relief, with a significant role played by the analgesic effects of platelet 5-HT [115].

## 3. The Contribution of PPP in PR-PRP

Blood plasma is a complex liquid base, protein-rich biological matrix in which platelets, leukocytes, and red blood cells are suspended [116]. Based on a quantitative proteomic analysis, we found that PPP contains plasma-based growth factors, a substantial amount of proteins, and cytokines [37].

### 3.1. Platelet-Poor Plasma Composition

PPP comprises biomolecules, extra-platelet growth factors IGF-1 and HGF [117], and many plasma proteins.

According to Anderson et al., blood plasma contains more than 1500 different proteins at varying concentrations [118], with 21 proteins accounting for approximately 99% of the total plasma protein composition [119]. As the plasma proteome is in direct contact with most cells in the body, it contains a broad variety of proteins from a wide range of intracellular locations and membrane proteins [119]. Plasma proteins can be coarsely classified into three groups: proteins present in high concentrations, “leakage” proteins that originate from pathological tissues [118], and several cytokines involved in immune disorders and inflammation [120].

The total plasma protein concentration has been determined to range from 60 to 80 mg/mL, with albumin (~60%), globulins (~35%), and fibrinogen (~4%) as three major protein groups. The remaining 1% of blood proteins represent several thousands of low-abundant proteins [116,121]. Table 2 outlines the most abundant human proteins, as well as their respective plasma concentrations and molecular weights.

The utilization of specific concentrated plasma proteins has gained interest in biotechnology, regenerative medicine, and orthobiological applications [29]. More specifically, interest is in the use of albumin in bioengineering, the employment of alpha-2-macroglobulins (A2Ms) to function as a protease inhibitor in osteoarthritis, and the employment of fibrin as a scaffold consolidated with other biologics, with emphasis on sustained cell release and tissue repair.

#### 3.1.1. Plasma-Based Growth Factors

PPP comprises plasma-based growth factors with active roles in several tissue-repair and -regeneration processes [123,124]. IGF-1 and HGF are plasma-based growth factors transported by several plasma proteins and are more prevalent in PPP when compared to PRP [117].

HGF is a multifunctional protein, primarily synthesized by hepatocytes, and circulates abundantly in plasma upon cell migration [125]. HGF has a relative molecular mass of 82 kDa and is a heterodimer molecule made up of an alpha unit and a beta unit [126]. HGF, secreted by fibroblasts and expressed in MSCs, affects epithelial cell proliferation, cell morphology, epithelial cell motility [127], and can induce epithelial tube formation [125,128].

HGF possesses mitogenic, morphogenic, and motogenic activities by interacting with the HGF/c-Met axis [126,129]. Additionally, HGF is a potent proliferative factor critical in angiogenesis [130]. Furthermore, HGF stimulates the proliferation and migration of progenitor cells and plays an important role in tissue regeneration [126,127], including skeletal muscle repair following injury [131].

Synergistic effects between HGF and VEGF have been reported [132], contributing to an increase in neovascularization with amplification of the angiogenic response as HGF stimulates the release of VEGF-A [39,48,133].

IGF-1 is a small peptide (7.6 kDa) with a molecular structure similar to insulin, displaying both anabolic and catabolic effects in many cells and tissues [124]. About 98% of the circulating IGF-1 in humans is bound to IGF-binding proteins (IGFBPs), with over 90% bound to IGFBP-3 [134], with a molecular weight range of 39 to 53 kDa. IGFBPs can regulate the activities of IGF-1 by reducing the free IGF-1 fractions and by prolonging its half-life time [135]. Importantly, IGF-1 bound to its specific binding proteins prevents passing through the hollow fiber pores of the ultrafiltration device, keeping IGF-1 in the final protein-rich PPP.

IGF-1 is responsible for cell growth and differentiation, as well as cell survival, protein synthesis, cell motility and proliferation, and DNA and matrix synthesis, particularly collagen-I [136].

IGF-1 has a high binding affinity for specific IGF-1 receptors and, to a lesser extent, for insulin receptors [137,138]. IGF-1 receptor binding activates PI3 kinase/Akt [139], the Ras-MAPK [140], and the PLC pathways [141], which are important in cell survival and cell signaling. Particularly in tendons, Ras-MAPK pathways regulate tenocyte proliferation and collagen synthesis [142]. Other IGF-1 receptor pathways are key mediators for several PGF, such as EGF, FGF, and PDGF [143].

There are three IGF-1 isoforms described: IGF-1Ea, IGF-1Eb, and IGF-1Ec [144]. The later isoform is a highly force-sensitive mechano-growth factor [145] and incites bone marrow MSC migration [146], stimulating tendon repair due to potent cell proliferative mechanisms. Potent IGF-1 anabolic effects have been demonstrated in chondrocyte regeneration in a dose-dependent manner [147]. Further, IGF-1 acts synergistically with platelet anabolic TGF-β1 growth factors [148].

Besides the IGF-1 anabolic effects, it is noticed that the biological IGF-1 activities lead to decreased catabolic events like, inhibiting degradative ECM effects by downgrading MMP-1, MMP-3, MMP-8, and MMP-13; creating a significant reduction in GAG release; blocking collagen from being released from the ECM; and inhibiting apoptosis [149,150]. Additional IGF-1 anabolic and catabolic mechanisms were mentioned in skeletal muscle pathways [151].

Importantly, endocrine, autocrine, and paracrine IGF-1 mechanisms stimulate angiogenesis by regulating some of VEGF’s activities by promoting vascular EC migration and tube formation [152,153].

IGF-1 is not found in significant quantities in platelet granules, as demonstrated in human and equine studies [154,155]. Therefore, mixing IGF-1 prepared from PPP with PRP-based TGF-β makes a strong case as a novel therapeutic biological preparation to treat OA, cartilage, tendon, and muscle pathologies. Hence, IGF-1 combined with PRP constituents might be considered a new effective biological product that can create both anabolic and catabolic tissue environments and potentially optimize inhibitory degradative pathways, induce MSC migration, stimulate chondrogenesis, and regulate apoptosis [156,157].

#### 3.1.2. Human Albumin

Human plasma albumin, produced in the liver, makes up about 60% of the total protein content in plasma, with a plasma concentration of approximately 40 g/L and a molecular mass of 66 kDa [158]. Its main function is to maintain the colloid osmotic pressure of plasma and can bind and transport fatty acids, hormones, and other molecules in the circulation [159]. Albumin is omnipresent in regenerative medicine research and bioengineering applications because of its structure and protein properties [160]. The multifaceted physiological roles of albumin include the recruitment by endogenous stem cells, promoting bone growth and anti-bacterial properties, and albumin can act as a free radical scavenger [161,162]. These typical albumin properties resulted in an omnipresence in regenerative medicine research and bioengineering applications in tissue matrix remodeling [163].

#### 3.1.3. Alpha-2-Macroglobulin

A2M is one of the largest plasma proteins in the human body, produced in the liver and released into the bloodstream [164]. The A2M glycoprotein is a homo-tetrameric glycoprotein containing four identical subunits, each with a molecular mass of 190 kD and an average plasma concentration of 1.4 g/L [122], whereas healthy synovial fluid contains approximately one-tenth of the plasma concentration [165].

A2M proteins inhibit proteolytic proteases that are known to be harmful to cartilage. In their mechanism of action, they utilize a tetrameric cage that physically surrounds a broad range of catalytic proteinases, a mechanism known as the protease snap-trap mechanism [166,167]. When large substrate molecules are trapped in the cage, they trigger A2M activation (a-A2M), which results in conformational protein changes [168]. Furthermore, in addition to capturing proteases, a-A2M exposes a reactive thioester, which forms covalent a-A2M/protease complexes with small primary amines [169,170].

The presence of A2M in synovial fluid has been postulated to inhibit various types of proteinases harmful to cartilage, possibly reducing OA symptoms and encouraging cartilage formation [171]. Platelet growth factors TGF-β1, FGF-2, macrophage activation factors, and TNF-α have a high binding affinity for aA2M [172,173,174,175]. Additionally, A2M captures and inhibits activated MMPs via the creation of A2M/MMP complexes and traps IL1β, thus reducing chondrocyte collagenase upregulation [176,177,178]. In a similar manner, A2M inhibits the activity of disintegrin and metalloproteinases with thrombospondin motifs (ADAMTS)-1,4,5,7, and 12, which contribute to the degradation of the ECM and OA [179].

The CORE™ ultrafiltration device (EmCyte Corporation^®^, Fort Myers, FL, USA) utilizes hollow fibers with a molecular weight cutoff of 20 kDa. Data from a case study revealed that the device efficiently concentrates plasma proteins, including A2M, when an average of 23 mL PPP yielded 6.6 mL of protein concentrate (Personal communications Dr. N. Stephens, director Biofyl Scientific Research, Fort Myers, FL, USA). Furthermore, an enzyme-linked immunoassay test determined that A2M concentrations ranged from 3800 to 6700 µg/mL, with a capture rate of 89.7%, which was approximately three times higher than the A2M concentration in the non-concentrated PPP volume. Early clinical case studies indicate that concentrated A2M supplementation may be beneficial in a number of MSK disorders, in particular in OA [180,181]. Further clinical studies are warranted and should be directed toward a better understanding of concentrated A2M dosing and immunomodulatory capacities related to chronic nociceptive inflammatory cartilage pathologies.

#### 3.1.4. Fibrinogen

Fibrinogen is a complex glycoprotein abundantly present in plasma at high concentrations, ranging from 1.5 to 4 g/L, with varying chain molecular masses (see Table 2) [182]. Fibrinogen consists of three pairs of non-identical polypeptide chains (α, β, γ) that are jointly connected by disulfide bridges to form two symmetrical half molecules [183]. The six fibrinogen polypeptides are arranged with N-termini in a central E-Domain and two outer D-domains via C-termini, with both domains connected by coiled-coil regions [184].

Fibrinogen is a soluble acute-phase protein macromolecule and acts as a precursor for fibrin formation when thrombin cleavages fibrinogen into an insoluble three-dimensional fibrin network, which is then cross-linked to form fibrin clots [185]. Furthermore, thrombin-mediated proteolytic cleavage activates platelets and supports invading macrophages, fibroblasts, ECs, and other molecules by entrapping them in the fibrin network [182,186]. Therefore, fibrinogen plays fundamental roles in hemostatic and homeostasis physiological processes to control bleeding, promote tissue repair, and enhance wound healing [187,188]. Additionally, fibrin(ogen) matrices are implicated in antimicrobial host defense mechanisms as they prevent microbial invasion by entrapping bacterial invaders and the recruitment of leukocytes [189], undertaking immunoregulatory functions through cell receptors of macrophages, neutrophils, and ECs [190,191,192].

## 4. PR-PRP Characteristics

PR-PRP is a liquid autologous product derived from a unit of whole blood following PRP processing. Typically, the PPP fractions are discarded after PRP processing; however, when preparing PR-PRP, the PPP fraction is processed with a concentrating device. A new generation biocompatible ultrafiltration device has been developed to concentrate the PPP proteins by processing this volume through hollow fiber semipermeable ultrafiltration [193]. By reducing plasma water in a controlled manner, a volume of PPP plasma proteins with a molecular weight larger than the ultrafiltrate membrane pore size is concentrated, along with several important extra-platelet growth factors, to a low-volume, viscous, protein-rich plasma product [194].

The highly concentrated PRP fraction and other cells are re-suspended in a low volume of plasma and extracted from the PRP device (Figure 2).

There are many commercial PRP devices utilizing either one-step or two-step centrifugation and preparation protocols to produce platelet concentrates, with many different bioformulations reported [195,196]. PRP device and preparation variable factors include the whole blood PRP processing volume, the prepared PRP volume, the presence of leukocyte, and the device platelet capture rate, affecting the PRP platelet concentration and platelet dose, as indicated in Table 3. When low-concentration PRP devices are used to create PR-PRPs, the cell concentration and the number of released matrices bound cells per time unit will be even lower, hence limiting the regenerative potential of the matrix.

A PR-PRP comprises a small volume of highly concentrated plasma proteins compounded with a low volume of highly concentrated PRP. To prevent preparations with low platelet concentrations embedded in the final PR-PRP matrix, ideally, PRP preparations with high platelet concentrations and platelet doses should be considered for PR-PRP preparations. Higher platelet and leukocyte concentrations initiate significantly greater cell proliferative, signaling, and angiogenetic activities during tissue repair and regeneration compared to PRP with low cell concentrations [6,36,39]. Dissimilarities in embedded matrix cellular platelet and leukocyte concentrations, as well as the total available platelet numbers for LP-PRP, LR-PRP, and PRF matrices, are indicated in Table 4 and visualized in Figure 3.

### 4.1. PR-PRP Matrix Formation

During tissue repair, regeneration, and wound healing, fibrin and platelet clot formation are integral parts of the natural healing process, as well as angiogenetic cascades [206] through intrinsic, extrinsic, and common pathways [207].

The formation of an engineered PR-PRP clot is very similar to ex vivo fibrin clot formation protocols, which involve moving cuvettes around their vertical axis at 37 °C to form a fibrin clot [208]. After the preparation of PRP, the PPP volume is processed in the ultrafiltration device, removing water from the PPP fraction and significantly reducing the PPP volume while simultaneously concentrating plasma proteins, plasma-based growth factors, and other molecules. The low-volume concentrated PPP fraction is then gently mixed and agitated with a low-volume, high-concentration PRP volume. The one syringe of consolidated soluble PRP and concentrated PPP fractions is labeled PR-PRP. A calculated volume of CaCl 10% (Ca^++^) is added to the PR-PRP volume, resulting in the conversion of prothrombin to thrombin. A complex multistep clotting process is then instigated and mediated by thrombin. Fibrin formation is initiated using the serine protease thrombin, which cleaves fibrinopeptides A and B (FpA, FpB) from fibrinogen [184]. The cleavage of FpA occurs first, inducing the polymerization of soluble fibrin monomers into protofibrils of half-staggered overlapping fibrin units. Thereafter, the released FpB is interrelated with the lateral aggregation of protofibrils, contributing to the clot tensile strength and fiber thickness.

In the next phase, as thrombin and Ca^++^ ions cleave the activation peptide from FXIII-a, a cross-linked three-dimensional insoluble fibrin network is formed, initiated by FXIII [209]. During the fibrin-polymerization process, covalent bonds are formed between fibrin γ-γ chains of the D domains, the γ and α-chains of the E-domain, and other plasma molecules [210]. Furthermore, at pathological tissue sites, the delivered PR-PRP will be exposed to factor III, also known as tissue factor (TF), a membrane-bound protein that will trigger thrombin generation and, ultimately, fibrin polymerization [211].

After fibrin polymerization via FXIII, the compact provisional three-dimensional PR-PRP matrix includes high concentrations of platelets trapped within the fibrin scaffold. Leukocytes and RBCs may also be embedded in the matrix depending on the formulation of the prepared PRP (see Figure 4).

### 4.2. PR-PRP Matrix Fibrinolysis

Fibrinolysis is an enduring physiological process of the biodegradation of fibrin clots and is characterized by a well-regulated balance between fibrin formation and breakdown [212]. This complex and dynamic process is mediated by plasmin, a serine fibrinolytic protease, which is obtained through the activation of the inactive glycoprotein plasminogen [213].

The cross-linked fibrin matrix promotes the rapid conversion of plasminogen via tissue-type plasminogen (t-PA) or urokinase-type plasminogen (u-PA) surface activators to the catalytically active protease plasmin [214]. tPA is the most abundant protease and acts primarily on fibrinolysis, while u-PA primarily targets neutrophils and macrophages’ cell surfaces for purposes such as ECM remodeling and monocyte migration [215]. Subsequently, plasmin initiates plasma hydrolysis of the matrix fibrin polymers [182].

In response to the proteolytic plasmin generation, plasma protein α2-antiplasmin and plasminogen activator inhibitor-1(PAI-1) are two foremost inhibitors of fibrinolysis acting as covalent inhibitors of plasmin to prevent plasminogen to bind to fibrin and to regulate the dissolution of fibrin polymers into soluble fragments, such as fibrin degradation products (FDP) and D-dimers [216,217].

In addition, after PPP ultrafiltration, high fibrinogen concentrations provide a denser fibrin matrix structure. Denser fibrin matrices have smaller pores and hence are less susceptible to fibrinolysis. A lower vulnerability can be attributed to the fact that smaller matrix pores inhibit plasminogen from binding to tPA surface activators [218,219]. Therefore, PR-PRP matrix biological components are released more slowly than in thin, non-concentrated fiber matrices with larger pores, leading to longer cell signaling and extended tissue repair processes [220].

### 4.3. Sustained Cellular Matrix Release

Upon application to pathological tissue and injury sites, soluble PR-PRP will become a fibrin-based matrix with typical mechanical properties and will exhibit several non-fibrin-based biological properties due to the embedded platelet and plasma-based growth factors, ECM proteins (fibronectin, vitronectin), and enzymes (plasminogen, tissue plasminogen activator) [221]. Furthermore, the provisional three-dimensional PR-PRP matrix serves as an environment for resident cells, homing MSCs, cytokines, and other cells that are involved in tissue remodeling, cell signaling, migration, proliferation, and ultimately tissue repair [213,222,223].

As a result of the dense fibrin architecture of the PR-PRP matrix, the embedded PRP cellular content, the ECM, and local resident tissue cells interact in a multidirectional and synergistic manner, allowing platelets to release their granular content, inflammatory cells, and leukocytes in a slow and progressive manner, steering the complex mechanisms involved in angiogenesis, immunomodulation, and tissue repair.

Interestingly, following fibrinolysis, the generation of FDP employs pro-inflammatory and anti-inflammatory properties. Pro-inflammatory D-dimers and fibrin fragment E stimulate the production of IL-6, IL-8, TNF-α, IL-1β, and chemokines, activating neutrophils and monocytes [224]. Meanwhile, fibrin fragment Bβ15-42 expresses anti-inflammatory effects [225].

In studies using PRFs and PRF membranes, the provisional fibrin matrix has been shown to allow for the sustained release of platelets and molecules [223,226]. However, PR-PRP matrices, consisting of LP or LR-PRP, possess a much greater biological profile than PRF-like and test tube preparations, given that PR-PRPs contain higher platelet concentrations, as specified in Table 4. Furthermore, following PPP ultrafiltration, the PR-PRP matrix fibrinogen concentration is significantly increased. Consequently, this results in fibrinolysis resistance, facilitating a prolonged sustained cellular and molecular release period [227], ultimately contributing to an increased level of cellular activity and angiogenetic stimulation [228,229]. A positive effect of a prolonged sustained release period of matrix embedded PGFs was confirmed in an animal model, demonstrating enhanced angiogenesis [230].

Notably, the preservation of PGFs in fibrin scaffolds and the sustained release of matrix cells over a prolonged period of time resulted in significantly greater cumulative amounts of growth factors compared to the initial burst release of PRP [231].

## 5. PR-PRP Matrix Biological Properties

PR-PRP, with fibrinogen as its main component, functions as a temporary three-dimensional scaffold. Fibrinogen will be converted into fibrin, and the PR-PRP cellular components will be embedded in the matrix. Following PR-PRP administration, the activated matrix demonstrates two main functions. First, the matrix displays mechanical properties, mimicking the ECM, and is able to fill partial and full-thickness tissue tears, therefore preventing the leakage of reparative cells in the affected area. Second, the matrix provides a molecular link for local tissue-resident cells and ECs to connect and invade the matrix microenvironment, mediating cellular interactions and signaling pathways [232] (Figure 5).

### 5.1. Fibrin, Fibrinogen, and Macrophage Responses

PR-PRP and other fibrin-based matrices are not only capable of providing scaffolds into which cells can be embedded or cells can be infiltrated, but they are also capable of direct molecular cell signaling, as these biological matrixes contain multiple molecular binding sites for growth factors, integrins, and other ECM molecules [233].

The provisional fibrin extracellular matrix provides a structural scaffold for cell infiltration and matrix anchoring at the tissue site, with a dynamic role for monocytes and macrophages in advancing and resolving inflammation in response to cues in their microenvironment [234].

Delgado et al. investigated the mechanical properties and cell behavior of three different fibrin-based matrix formulations, including a matrix with a high fibrinogen content, similar to the concentrated PPP following ultrafiltration [235]. Interestingly, a fibrinogen-concentration-dependent cell adhesion was observed, which was confirmed by others [236,237]. It was concluded that high fibrinogen concentrations improved the matrix mechanical properties, with better cushioning properties and cell-adhesion capacity without impeding the cell viability due to a longer sustained release period. Further, the improved cushioning effects contributed to better chondrogenic properties, and it was suggested that high-density matrices might be suitable for the treatment of chondral defects [238].

Importantly, the utilization of high fibrinogen content matrices did not alter cell properties or viability due to swelling, and a normal nutrient-diffusion capacity was observed. Additionally, no decrease or inhibition of cell migration and proliferation was seen [239].

There is extensive knowledge about how soluble factors such as cytokines and chemokines regulate macrophage polarization [240,241]. However, the effects of the insoluble fibrin matrix on macrophage behavior are not well understood. However, it has been demonstrated that fibrin(ogen) can modulate the activities of monocytes and macrophages and play an important role in the transition from inflammation to tissue repair [242].

Hsieh et al. used cultured macrophages and activated fibrinogen with thrombin to produce a fibrin gel [243]. They concluded that cultured macrophages on fibrin gels secrete the anti-inflammatory cytokine IL-10, suggesting that fibrin can promote an anti-inflammatory response in macrophages, even when stimulated by IFN-γ.

### 5.2. Fibrin to Support in Immune Responses and Inflammation

Immune cells contain an arsenal of fibrinolytic regulators and plasmin and participate in host immune responses, providing access for immune cells to migrate to the dissolving matrix and FDP. In turn, the fibrinolytic proteins have diverse roles in immunoregulation [244].

Macrophage-1 antigen, a surface receptor on monocytes, macrophages, and neutrophils, contributes to the adherence to ECM proteins, regulates adhesion-dependent processes in leukocytes, and functions as well as an important fibrinogen receptor [245]. Porous fibrinogen 3D scaffolds led to resolving local inflammatory responses in a bone fracture model, measured by decreased pro-inflammatory cytokines, like IL-8, IL-6, IL-17, IL-1b, and TNF-α. It was concluded that the fibrinogen scaffolds provided a temporary support matrix to promote a pro-regenerative local environment [246]. Almeida et al. investigated the role of fibrinogen in immunomodulation and concluded that fibrinogen instigated immune cell responses, with increased recruitment of MSCs and downregulation of pro-inflammatory molecules [247].

More specifically for PR-PRP matrix formulations, activated platelets shed platelet microparticles (PMPs) that exercise anti-inflammatory and immunosuppressive effects and down-regulate macrophage and dendritic cell activation to control inflammation and immune responses [248]. Additionally, PMPs can support resolving tissue inflammation by inhibiting IL-17 production by regulatory T cells [249], and PMPs can alter monocyte and macrophage polarization to less reactive phenotypes [250].

Noteworthy, Kim et al. demonstrated that three-dimensional fibrin-based scaffolds improve the paracrine effects of bone marrow-derived MSCs, resulting in elevated levels of immunomodulatory factors and function [251]. 

#### PR-PRP’s Clinical Application Potential

The use of autologously prepared protein-rich fibrin matrices is widespread in clinical procedures and biomolecular engineering applications, in which they have been recognized as versatile carriers for many cell types and biomaterials [252]. They can be used in non-surgical tissue repair and regeneration procedures as a standalone injectable preparation or as a cellular remodellable tissue adhesive [253,254]. Furthermore, fibrin matrices can be safely and effectively used as fibrin sealants in biosurgical procedures to improve patient outcomes [255,256]. Table 5 illustrates the multiple roles and clinical potential of fibrin matrix utilization.

### 5.3. Endothelial Cell Interactions with Fibrin Matrix

Fibrin matrices are portrayed as an excellent matrix for the invasion and adhesion of ECs, followed by the formation of new capillary-like structures [257]. Fibrin stabilizes the expression of vitronectin receptor avβ3-integrin on ECs, promoting their migration to the matrix proteins [258] and several integrins and other receptors, like ICAM-1 and VE-Cadherin 1, facilitate the binding of ECs to the fibrin(ogen) and leukocytes [259].

After PR-PRP delivery and the subsequent fibrin, protein, and activation, leukocytes infiltrate into the developed matrix, followed by ECs from adjacent tissues [260]. Matrix-embedded ECs are associated with angiogenesis and tissue-repair processes as they realign with new vascular structures [233]. Furthermore, encapsulated ECs produce and secrete t-PA and u-PA, converting plasminogen into plasmin and initiating matrix fibrinolysis [261]. The ensuing proteolytic matrix components are found to be essential in prompting an early angiogenetic response [262]. FDP increases the release of endothelial cell-derived growth factors [263], and D-dimers induce the secretion of IL-1 from macrophages, stimulating the expression of PAI-1 in ECs [264].

### 5.4. Role of Fibrin Matrix in Angiogenesis

Fibrin-based matrices and angiogenesis are closely connected to physiological processes. Restoring the microcirculation to its former state can be supported by PRP utilization and is determined by the release of PGFs, pro- and anti-angiogenic factors, and EC responses [39]. However, fibrin matrix-based angiogenesis is driven by biochemical mechanisms, the matrix milieu and mass, and the attachment of ECs to the matrix [265]. Similarly, fibrin induces the expression of αvβ3 integrin, an ECM cell-surface receptor, which allows ECs to bind to fibrin itself and ECM proteins fibronectin and vitronectin, promoting angiogenesis and wound repair [258].

As opposed to non-protein concentrated fibrin matrixes with a low platelet concentration, PR-PRP matrix formulations contribute significantly to angiogenic mechanisms and tissue repair. As part of the PR-PRP matrix formulation, high-definition PRP is an integral component, embedding high levels of PGFs, angiogenetic factors, and other molecules in a dense three-dimensional matrix structure. Furthermore, angiomorphogenic effects and capillary morphogenesis are dependent on the availability of PRP growth factor concentrations, as well as the matrix rigidity and structure [266].

### 5.5. Antimicrobial Activities of Fibrin(Ogen)

In addition to its role in hemostasis, fibrinogen plays an important role in host defense against microorganisms. In response to pathogens and their environment, the fibrin matrix may trigger a protective immune response with antimicrobial properties mediated by fibrin(ogen) through two distinct mechanisms [189]. Firstly, the fibrin matrix acts as a protective barrier that physically entraps bacteria or encapsulates foci of bacteria to limit their growth and dissemination. Secondly, fibrin can stimulate immune cell recruitment and activation, which results in the elimination of pathogenic microbes.

The non-exhaustive Table 5 displays a variety of biological characteristics and therapeutic goals of fibrin(ogen) and compound PR-PRP matrixes. Fibrin-based matrices have distinct therapeutic objectives based on their biological characteristics. In addition to sharing many biological functions and application objectives with fibrin-based matrices, PR-PRP matrixes provide the additional benefit of embedding the cellularity of PRP formulations directly within the matrix, thereby allowing a broader range of therapeutic applications and treatment objectives.

It has been demonstrated in several studies that multiple bacterial species are susceptible to host fibrinogen molecules, including neutrophils and macrophages, when exposed to fibrin deposits [267,268]. Conversely, pathogen clearance in mice lacking fibrinogen was significantly compromised, suggesting a failure in innate immune cell function by fibrinogen integrin Mac-1 [269].

### 5.6. Matrix Similarities and Differences

Based on their respective cellular components, fibrin(ogen) and PR-PRP matrixes display similar biological and therapeutic characteristics, along with distinct differences, as indicated in Table 5. Autologous PR-PRP and fibrin matrices, such as PRF, PRFM, and Vivostat™ technology [270,271,272], are prepared from the patient’s whole blood and are highly regarded for their safety and versatility. One of the key properties of autologous PR-PRP and fibrin matrices is their rapid polymerization, enabling them to be utilized as a fibrin tissue sealant or adhesive, depending on specific therapeutic goals. Both PR-PRP treatment modalities possess the same biological functions as the fibrin(ogen)-based approaches, albeit with a few notable differences. A key difference between the PR-PRP matrix and the fibrin matrix is the presence of concentrated fibrinogen in the PR-PRP matrix. This biocomponent contributes to a denser sealant and matrix, making it more resistant to fibrinolysis. Additionally, it promotes a prolonged sustained cellular release phenomenon. The second important difference between the two matrices is the use of high PRP concentrations in the PR-PRP matrix. Whether with or without specific leukocytes, these high platelet concentrations guarantee a significant amount of additional physiological and biological functions that are unique for platelets and leukocytes, allowing for a surplus of repair and regenerative, synergistically functioning pathways. By leveraging these pathways, the sealant and matrix PR-PRP modalities enhance the overall regenerative potential of the treatment while offering a broader range of effective treatment options for patients.

## 6. Limitations and Future Directions

Fibrin matrices with embedded platelets and other cellular concentrations have favorable features that allow them to act as biological preparations. However, poor fibrin matrix properties can occur with reduced tissue-adhesion capacities as a result of low fibrinogen concentrations, limiting the sustained release period when a prolonged matrix-to-tissue adhesion period is required. A higher concentration of fibrinogen and an increase in cross-linked fibrin fibers could improve the matrix’s mechanical properties.

Furthermore, depending on the employed commercially available PRP preparation methods, differences in embedded total platelet content and bioformulations can occur, releasing varying cellular and biomolecular numbers during fibrinolysis.

Future directions should include clinical and in vitro studies to evaluate the duration of the sustained-release mechanism, matrix tissue activation, and, finally, clinical effectiveness when compared to single, non-compounded biological treatments. However, the application of PR-PRP matrices will pave the way to investigate new treatment options in patients with complex pathological conditions or who are not candidates for surgical repair.

## 7. Conclusions

In our opinion, fibrin-based matrices have emerged as a promising biological and multipurpose platform due to their high cellular-adhesion capacity.

Protein-rich autologous platelet concentrates and their matrixes are complex multidimensional autologous biological products that are a combination of a high concentration of platelets and a high concentration of proteins.

High-volume 2-spin PRP devices employing variable biological-preparation methods are well-suited for preparing different PR-PRP formulations. These devices effectively capture high platelet numbers, accommodate variations in bioformulations, and produce sufficient volumes of PPP to concentrate plasma proteins, in particular A2M and fibrinogen.

After activation, the PR-PRP protein network, in particular fibrinogen, is enzymatically converted into fibrin, and the liquid PR-PRP will transform into an insoluble and elastic three-dimensional porous matrix. The concentrated fibrin(ogen) provides the matrix with additional strength, shape, and stability when compared to non-concentrated fibrinogen matrices.

The PR-PRP matrix is embedded with platelets and other cells, creating a favorable environment for cellular interactions and cell signaling. Furthermore, matrix cell receptors connect and bind with invading MSCs, macrophages, fibroblasts, and ECM proteins. As a result of naturally occurring biological fibrinolysis, matrix-ingrained platelets and other biomolecules are released over time, ensuring a progressive delivery of cells and biomolecules to the local, pathological microenvironment. Synergistic matrix properties are initiated in the presence of high concentrations of reparative matrix cells, which enhance cell signaling and promote reparative activities, including cell migration and proliferation, immunomodulation, angiogenesis, and inflammation control, thereby supporting and initiating reparative and regenerative effects.

In our PR-PRP preparations, we obtained 52 mL of autologous blood to produce 3 mL of high-concentration PRP and 3 mL of concentrated protein-rich PPP, creating a 6 mL biological active matrix for tissue repair, regeneration, and wound healing. To determine the biological impact of PR-PRP matrix formulations on patient outcomes, more studies are required comparing PR-PRP matrices, non-concentrated fibrinogen matrix formulations, and non-matrix-based biological preparations.

## Figures and Tables

**Figure 1 ijms-25-07914-f001:**
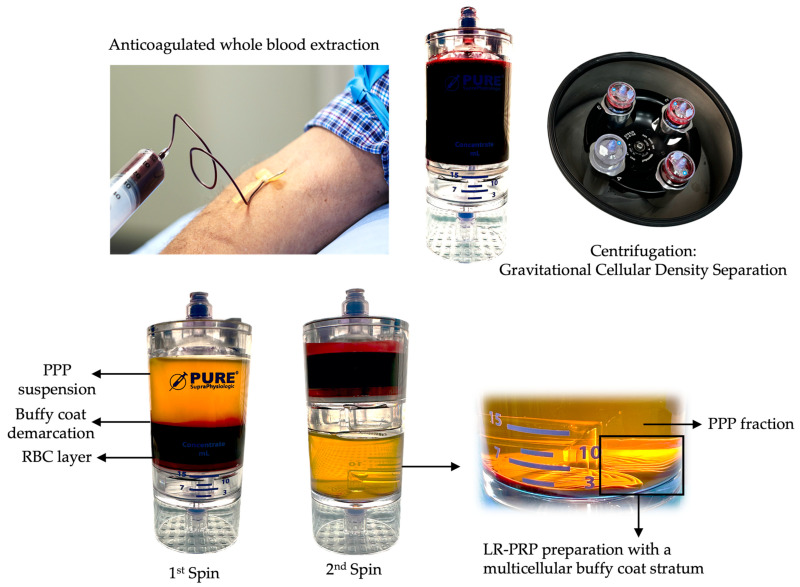
Portrayal of a 2-spin centrifugation method used to produce autologous PRP. PRP preparation involves the collection of a predetermined volume of peripheral blood in collection syringes containing an anticoagulant, like calcium citrate 3.8%. The predonated whole blood is gently loaded in a PRP device for gravitational cellular density separation using a two-spin centrifugation protocol. After the first spin cycle, the whole blood components are separated into three basic layers: the PPP suspension, the buffy coat demarcation layer, and the RBC layer. During the second centrifugation cycle, the platelet and other cells in the PPP fraction and RBC layer are further separated, resulting in a multicellular buffy coat stratum containing high concentrations of platelets and, eventually, leukocytes. A calculated portion of the PPP fraction is removed, leaving the platelet concentrate within a small volume of plasma for platelet resuspension. Thereafter, the PRP is extracted from the device. In this graphic, LR-PRP has been prepared, and the multicellular fractions consist of a high concentration of platelets, monocytes, lymphocytes, neutrophils, and some red blood cells. Abbreviations: PPP: platelet-poor plasma; RBC: red blood cells; LR-PRP: leukocyte-rich PRP; PRP: platelet-rich plasma.

**Figure 2 ijms-25-07914-f002:**
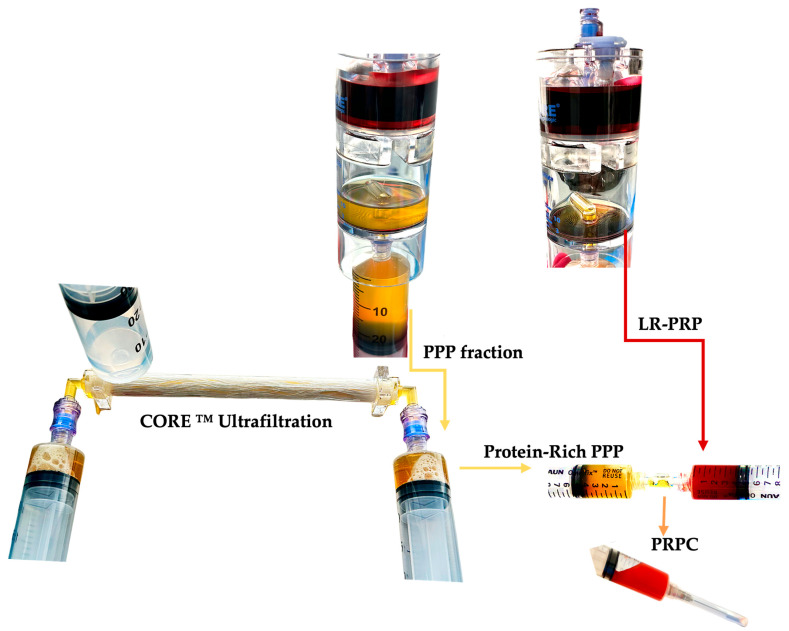
PR-PRP preparation. After the second spin, a calculated portion of the PPP fraction is removed with a syringe and attached with an empty syringe to the ultrafiltration device, as well as an effluent collection syringe, to collect the eliminated plasma water. The PPP syringes are manually pushed through the device, and plasma water, proteins smaller than 20 kDa, and cytokines are removed through the hollow fiber pores. After a series of passes, the PPP volume is significantly reduced, leading to a small and viscous volume of protein-rich plasma. The remaining PPP volume in the PRP device is used to resuspend the highly concentrated multicellular LR-PRP fraction and capture it from the PRP device. The concentrated protein-rich PPP and LR-PRP are consolidated into one syringe and gently mixed, creating PR-PRP. Abbreviations: LR-PRP: leukocyte-rich PRP; PPP: platelet-poor plasma; PRP: platelet-rich plasma; PR-PRP: protein-rich platelet concentrate. (the ultrafiltration device shown is the CORE™ Ultrafiltration System, developed by EmCyte Corporation^®^, Fort Myers, FL, USA).

**Figure 3 ijms-25-07914-f003:**
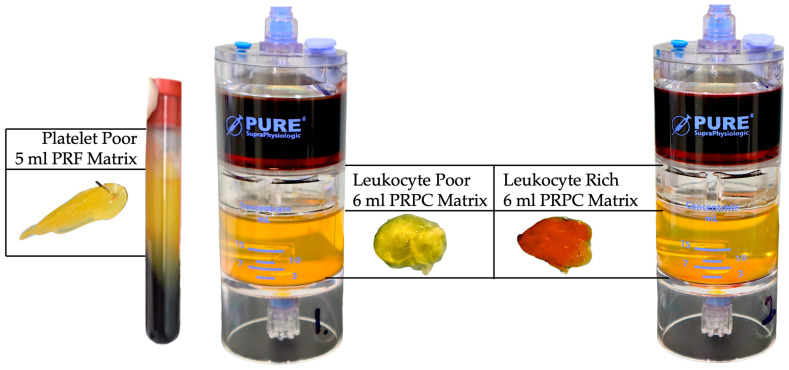
Visualization of PRF, LP-PR-PRP, and LR-PRP PR-PRP matrices. The PRF matrix and leukocyte-poor and leukocyte-rich PR-PRP matrices are exhibited. In a same-patient experiment, 130 mL of anticoagulated whole blood was extracted to prepare 10 mL of PRF, LP-PPR, and LR-PRP, using two 60 mL 2-spin PRP devices (PurePRP^®^SP, EmCyte Corporation^®^, Fort Myers, FL, USA). After the first spin, the PRF clot was removed from the test tube. After the 2nd spin of the LP and LR-PRP preparations, the PPP fraction was removed for ultrafiltration to concentrate the plasma to produce protein-rich PPP resuspended with the concentrated PRP fractions, described in detail in Figure 2. Exactly 3 mL of both PRP formulations was consolidated with 3 mL of protein-rich PPP. To create the PR-PRP matrices, normal baseline clotting parameters were restored by adding 0.35 mL of NaCl 10% to the 6 mL PR-PRP volume. Thereafter, 150 IU of bovine thrombin was added to the re-calcified PR-PRP volume, mimicking the effect of TF for matrix formation.

**Figure 4 ijms-25-07914-f004:**
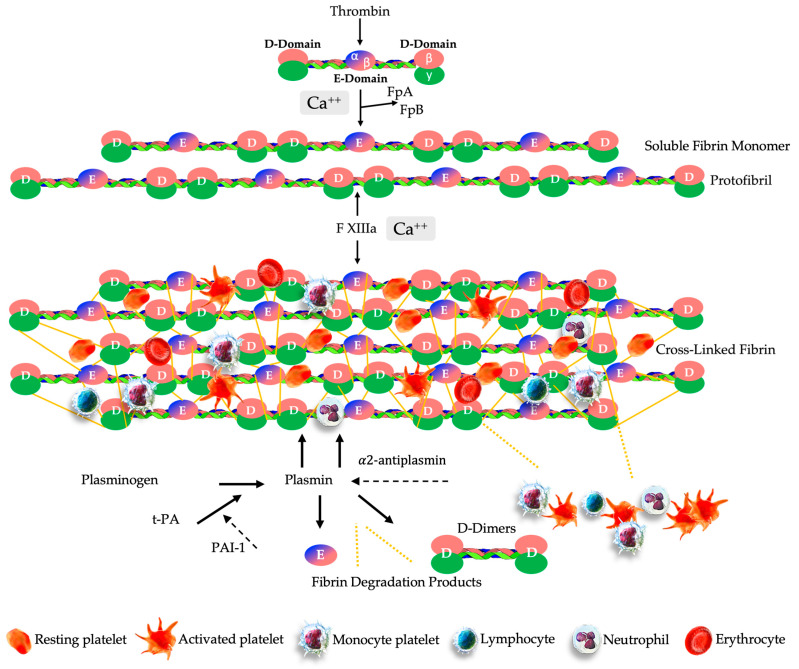
Schematic overview of the multistep physiological process of PR-PRP matrix creation and the sustained release of matrix biological components. The ultrafiltration device concentrates non-activated plasma proteins, including fibrinogen, consisting of two D-domains composed of α, β, γ chains, and plasma growth factors. After consolidating a small volume of high-concentration PRP with the concentrated protein suspension, fibrinogen undergoes a structural change in the presence of thrombin and calcium ions, leading to the cleavage of fibrinogen into FpA and FpB to form stable, complex, soluble fibrin monomers. The soluble monomer polymerizes to form half-staggered protofibrils. Several enzymes, including FXIIIa, and in presence of Ca^++^ ions, protofibrils are converted to the cross-linked fibrin embedded with highly concentrated PRP cells. Fibrinolysis is initiated when t-PA converts plasminogen into plasmin, whereas PAI-1 inhibits t-PA, preventing the activation of plasminogen and thus fibrinolysis. Hence, FDP, D-dimers, platelets, and other cells are continuously released from the PR-PRP matrix, whereas α2 antiplasmin acts by blocking plasmin activity, reducing the proteolytic fibrinolytic breakdown.

**Figure 5 ijms-25-07914-f005:**
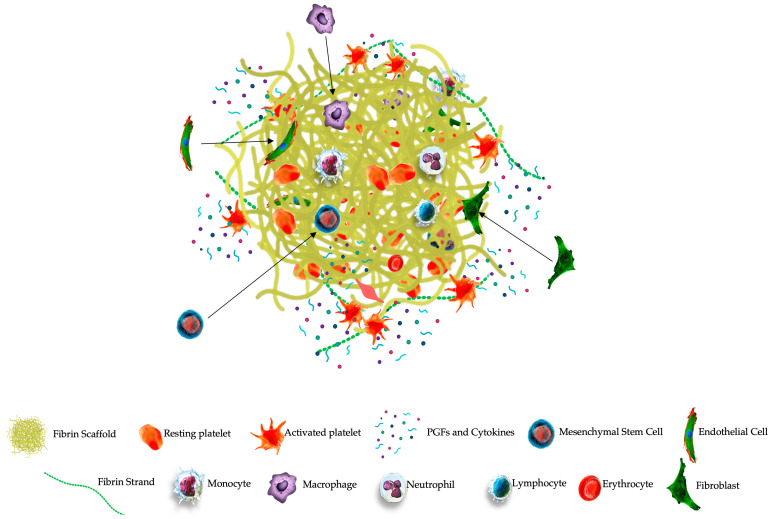
Illustration of the biological features of a PR-PRP matrix. The beneficial effects of an accurately prepared and applied PR-PRP matrix in pathological microenvironments are depicted by several biologically induced processes that are crucial in the repair and regeneration of diseased tissues. The PR-PRP matrix, which is composed of concentrated and activated fibrinogen as its main component, serves as a temporary dense insoluble three-dimensional scaffold with high concentrations of PRP cellular content embedded. Importantly, the matrix enhances the viability and functionality of the PRP cellular content. The PR-PRP matrix is instrumental in providing a molecular link for invading local tissue-resident cells, such as MSCs, macrophages, fibroblasts, and ECs. Inside the matrix microenvironment, a multitude of cells and molecules engage in many cellular interactions. Upon fibrinolytic breakdown of the PR-PRP matrix, an abundance of activated platelets, their PGFs, and other biologically active molecules and cells are released to the local microenvironment. Ultimately, the sustained release of matrix substances leads to an increase in cellular activity and signaling, angiogenetic and immunomodulatory processes, and antimicrobial activities, contributing to the overall tissue repair and regenerative process. Moreover, as a result of the breaking down of fibrin strands, they provide structural support for the development of new tissues and facilitate the adhesion and migration of cells [40,185,191,192,193,194,195].

**Table 1 ijms-25-07914-t001:** Partial list of PR-PRP components, structures, and molecular content of platelets and plasma.

PR-PRP Component	Structure	Key Content	Main Functions
Platelet	α-granules	Growth Factors: PDGF (AA-BB-AB-CC), VEGF, TGF (α-β), FGF (a-b), EGF, CTGF	Growth factor-based regulation of tissue repair via cell proliferation, differentiation, mitogenesis, chemotaxis, and epithelial repair.
		Adhesive Proteins: Fibronectin, vitronectin, fibrinogen, vWF, P-selectin, integrins αIIbβ, Phosphatidylserine	Platelet aggregation, platelet–endothelial cell interaction, and thrombus formation.
		Coagulation Factors: Factors IV, XI, XIII, plasminogen, plasmin, antithrombin, tissue factor	Hemostasis and thrombus formation.
		Angiogenic Regulators: IL8, thrombospondin, Angiostatin, PF-4, TIMP-1,4, MMP-1,2,9, Angiopoietin, Endostatin, SDF-1, PMP	Angiogenesis cascades and re-establishing vasculature.
		Cytokines: IL1, IL4, IL6, TFNα, SDF-1	Chemotaxis, inflammatory response modulation, and antimicrobial activity.
		Chemokines: RANTES, CXCL4, CXCL7, CCL2, CCL3, CCL5, β-TG	Inflammation, antimicrobial, and bactericidal activity.
		Complement Proteins: C3, C4	Phagocytosis, chemotaxis, and platelet activation.
		Exosomes: mRNA, miRNA, CXCL4, CXCL7	Cell adhesion, paracrine communication, regulation of cell fate, and modulation of inflammatory response.
	Dense granules	ADP, ATP, TFNα calcium, serotonin, epinephrine, pyrophosphates	Platelet activation and vasoconstriction.
	Lysosomes	Collagenase, elastase, Cathepsin, α-arabinoside, β-galactosidase	Matrix degradation and antimicrobial activity.
	Multivescicular Bodies	Exosomes Extracellular vesicles	Cell proliferation, PGF transportation, and platelet–cell communication.
PPP	Plasma Proteins (>300)	Albumen, fibrinogen, Alpha-2-macroglobulin,	Blood clotting, maintain blood pressure, carrier functions, immunity, and pH regulation
	Coagulation factors	Tissue factor, Factor I, II, IV, V, VII, vWF	Intrinsic and extrinsic coagulation pathways and clot formation
	Growth Factors	IGF-1, HGF	Bone growth, glucose transport in fat and muscle, muscle production, mitogenesis, cell growth, and cell proliferation

Abbreviations: PDGF: platelet-derived growth factors; VEGF: vascular endothelial growth factor; TGF: transforming growth factor; FGF: fibroblast growth factor; EGF: epidermal growth factor; CTGF: connective tissue growth factor; vWF: von Willebrand factor; IL: interleukin; PF-4: platelet factor 4; TIMP: tissue inhibitor of metalloproteinase; MMP: matrix metalloproteinase; PMP: platelet-derived microparticles; TNF: tumor necrosis factor; SDF: stromal cell-derived factor; RANTES: regulated by T-Cell activation and probably secreted by T-Cells; CXCL: CXC chemokine ligand; CCL: C-C motif chemokine ligand; C: complement protein; ADP: adenosine diphosphate; ATP: adenosine triphosphate; mRNA: messenger RNA; miRNA: microRNA.

**Table 2 ijms-25-07914-t002:** Overview of plasma proteins.

Plasma Protein Chains	Concentration (g/L)	Molecular Weight (kDa)
Albumin	40	66
IgG γ-chain	12	50
Transferrin	2.3	25
IgA α-chain	2	60
Apolipoprotein A1	1.4	28
α2-macroglobulin	1.4	190
α-1antitripsin	1.1	52
Fibrinogen α-chain	0.95	95
IgM µ chains	0.75	75
Hemopexin	0.75	60
Apolipoprotein B	0.72	250
α1-acid glycoprotein	0.61	41
Fibrinogen β-chain	0.56	56
Apolipoprotein AII	0.3	110
Fibrinogen γ-chain	0.5	50
Complement C3 β-chain	0.39	75
Antithrombin III	0.32	58
Apolipoprotein AII	0.3	17
Haptoglobin α-chain	0.29	40
Pre-albumin	0.26	16
Ceruloplasmin	0.21	132
Haptoglobin β-chain	0.14	20
Fibronectin	0.11	230
Plasminogen α-chain	0.099	60
Complement C4 α-chain	0.082	98
Complement C4 β-chain	0.061	73
Plasminogen β-chain	0.041	25
Complement C4 γ-chain	0.028	33
Other	0.038	N/A

An overview of the most abundantly present plasma proteins, their specific protein chains, concentrations, and molecular weight expressed in kiloDaltons (kDa). Adapted and modified from Perrin [122]. N/A: not applicable.

**Table 3 ijms-25-07914-t003:** Hematology data from different PRP preparations.

Device Category	PRPv mL	PLTc ×10^3^/µL	PLTd ×10^6^	WBCc ×10^3^/µL	MONc ×10^3^/µL	NEUc ×10^3^/µL	RBCc ×10^9^/µL	References
P-PRP	4.8	170	887	0.3	0	0	0	[197,198,199]
PRF	5	205	1.025	0.1	0	0	0	[195,200,201]
LP-PRP	4.6	1280	5.686	11.5	3.1	0.8	0.2	[111,202,203]
LR-PRP	6.3	1603	9.212	24.7	4.7	5.4	1.6	[202,204,205]

The data are extrapolated and calculated from relevant published studies to demonstrate the variances in platelet and leukocyte concentrations, as well as platelet dosages of four different PRP formulations. Abbreviations: PRPv: platelet-rich plasma volume; PLTc: platelet concentration; PLTd: platelet dose; WBCc: white blood cell concentration; MONc: monocyte concentration; NEUc: neutrophil concentration; RBCc: red blood cell concentration; P-PRP: pure platelet-rich plasma; PRF: platelet-rich fibrin; LP-PRP: leukocyte-poor PRP; LR-PRP: leukocyte-rich PRP.

**Table 4 ijms-25-07914-t004:** Indicative matrix cellular differences and total available platelets for 3 different PRP preparations.

Matrix Formulation	PLTc ×10^6^/µL	PLTs Available in Matrix, ×10^9^	MONc ×10^3^/µL	NEUc ×10^3^/µL	RBCc ×10^9^/µL
PRF	0.275	1.375	0	0	0
LP-PR-PRP	2.3	6.9	4.5	6.1	0.3
LR-PR-PRP	3.9	11.8	8.2	13.5	2.4

Three different matrix formulations are displayed, demonstrating the differences among them regarding platelet concentration, total available platelets, and leukocyte cell concentrations. The PRF matrix data are composed from the data in Table 3. The hematological data for PRF and both LP and LR-PR-PRP formulations originate from a same-donor experimental model. Both PR-PRP matrix formulations consisted of a mixture of 3 mL of PRP with 3 mL of protein-rich plasma to a consolidated volume of 6 mL. Abbreviations: PRF: platelet-rich fibrin; LP-PR-PRP: leukocyte-poor protein -rich PRP; LR-PR-PRP: leukocyte-rich protein-rich PRP PLTs: platelets; PLTc: platelet concentration; MONc: monocyte concentration; NEUc: neutrophil concentration; RBCc: red blood cell concentration.

**Table 5 ijms-25-07914-t005:** A variety of biological and therapeutic hallmarks of autologous fibrin(ogen) matrices and compound PR-PRP matrices.

Treatment Modality and Biocomponent	Biological Function of Matrix	Therapeutic Goals
Sealant/Glue Fibrin(ogen)	Post-surgical hemostasis control Oozing tissue sealing Controllable biodegradation	Avoid seromatous wound leakage Avoid blood loss Decrease surgical adhesions Periodontal membrane
	+	
Regenerative Glue Fibrin(ogen) and PRP	ECM cellular support Anti-inflammatory Active platelet-mediated angiogenesis	Support surgical tissue healing Prevent infection Chronic wound and bone healing Scar reduction
Fibrin Matrix Fibrin(ogen)	Temporary matrix for invading platelets and leukocytes EC migration Cell adhesion Promote angiogenesis process Collagen production Growth factor carrier Promote immunomodulation Fibrinolytic sustained release	Tissue repair Tissue regeneration Tissue engineering heart valves and patches Hydrogel transport Intra-osseous infiltration Nerve injuries Myogenesis
	+	
PR-PRP matrix Fibrin(ogen) and PRP	Osteoblast stimulation Temporary matrix with embedded concentrated platelets and leukocytes Platelet growth factor reservoir Plasma growth factor reservoir Tissue ingrowth stimulation Instigator immunomodulation Cell proliferation–differentiation Organizer angiogenesis process MSC and HF stem cell paracrine effects Antimicrobial pathways Nociception A2M enzymatic processes	Bone defect augmentation Epithelialization chronic wounds Tissue infection management Tissue repair, including nerves Tissue regeneration Skin grafting Wound healing Partial-Full thickness tendon/ligament biofiller Immunomodulation Osteoarthritis Joint protease inhibitor Joint cushioning Hair growth

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
