# Peer review of "Profound Properties of Protein-Rich, Platelet-Rich Plasma Matrices as Novel, Multi-Purpose Biological Platforms in Tissue Repair, Regeneration, and Wound Healing"

_ijms, 2024, doi:10.3390/ijms25147914_

Round 1

Reviewer 1 Report

Comments and Suggestions for Authors

Comments on the Quality of English Language

Minor editing of English language required

Author Response

Dear Reviewer 1.

Thanks for your time to review our manuscript. The authors appreciate your work and knowledge on the subject.

In general, we decided to modify the title to Protein Rich Platelet Rich Plasma (PR-PRP) instead of protein rich platelet concentrate, as we the authors believe this is a more representative and accurate definition according to the described PRP formulations described in this manuscript. PRPC has been changed in entire manuscript to PR-PRP. We hope you agree.

Based on your review we address here each comment separately and they are visible in track changes.

  1. The font type has been adjusted, thank you for this comment
  2. The latest classification attempt of Kon et al. has been mentioned and referenced.
  3. An additional column with relevant references in table 1 has been added, thank you for this suggestion.
  4. Subsection 2.5.1 is referencing Anti-inflammatory; we appreciate the suggestion. The subsequent sub chapters are re-numbered.
  5. We were not able to find any other similar ongoing study or published paper mentioning a similar matrix structure consisting of a high fibrinogen concentration, including plasma-based growth factors, with an embedded high definition PRP. Plenty full papers have been published using various platelet rich fibrin formulation and matrices, however they contain non-concentrated proteins and platelet formulations.
  6. Thank you for your comment regarding section 3, and we agree that the information provided is extensive, but we believe also complete. We modified this section slightly, as we believe that the organization of this section is represented in a manner that non-experienced readers are well informed.
  7. Based on your recommendation we added references to table 5. Thank you for this feedback.
  8. We added as per your suggestion a paragraph on limitations and future directions, thanks. Conclusions are now paragraph 7.
  9. We added several publications from others in the relevant positions as per your comment.

The authors thank  you for your appreciated feedback and suggestions and hope we have improved the overall manuscript.

Prof. Dr. Peter A . Everts

Corresponding author.

Reviewer 2 Report

Comments and Suggestions for Authors

I would like to commend the authors on their review, as it is always important to gather new works for clarification and new insights. It seems to me to be a comprehensive and clear paper that effectively outlines the role of PRP. One clarification that I feel compelled to mention relates to Figure 4, which is beautiful but could be simplified. I understand that it's a matrix, but increasing the distance between the D-dimers would better highlight the presence of the cells shown in the figure, as clearly stated in the caption.

Author Response

Dear Reviewer 2.

Thanks for your time to review our manuscript. The authors appreciate your work and knowledge on the subject.

In general, we decided to modify the title to Protein Rich Platelet Rich Plasma (PR-PRP) instead of protein rich platelet concentrate, as we the authors believe this is a more representative and accurate definition according to the described PRP formulations described in this manuscript. PRPC has been changed in entire manuscript to PR-PRP. We hope you agree.

Based on your review we slightly modified Figure 4, with regard to your comment on D-Dimers.

The authors thank you for your appreciated feedback and suggestions, and hope we have improved the overall manuscript based on your suggestion.

Reviewer 3 Report

Comments and Suggestions for Authors

The main aim and question of this manuscript is to provide a comprehensive overview of a multipurpose biological platform comprised of high-definition PRP and viable protein-rich plasma, as well as their typical biological contributions in tissue repair, regeneration, and wound healing.

Regarding the originality of this manuscript, this manuscript shows rich content, providing a deep insight for some works: the study is within the journal’s scope, and I found it to be well-written, providing sufficient information. The main topic is very original, and of great clinical impact. I found the manuscript to be well-written, with an organic overview, and a densely organized structure. The text is very clear and easy to read, but need some improvements. Even if the manuscript provides an organic overview, with a densely organized structure and based on well-synthetized evidence, there are some suggestions necessary to make the article complete and fully readable. For these reasons, the manuscript requires major changes.

Please find below an enumerated list of comments on my review of the manuscript:

INTRODUCTION:

LINE 59: Regenerative medicine is a discipline continuously in evolution, where platelets and dentinal derivates are used as autologous biomaterial, but also the experimental use of stem cells in vitro studies gave promising results (see, for reference: Patil, S.; Reda, R.; Boreak, N.; Taher, H.A.; Melha, A.A.; Albrakati, A.; Vinothkumar, T.S.; Mustafa, M.; Robaian, A.; Alroomy, R.; et al. Adipogenic Stimulation and Pyrrolidine Dithiocarbamate Induced Osteogenic Inhibition of Dental Pulp Stem Cells Is Countered by Cordycepin. J. Pers. Med. 202111, 915).

This is the sentence, which I refer:

“A wide range of medical applications use autologous platelet-rich plasma (PRP) formulations and plasma protein-based biological preparations for non-surgical tissue repair, regeneration, and wound healing applications (1,2)”.

In fact, before starting to describe the application of autologous PRP, in my opinion the manuscript may benefit from providing a very concise premise, about the evolution of regenerative medicine, based on the application of different classes of biomaterials (among the PRP), as suggested by recent scientific data. Anyway, it is not important the presence of the line itself, but for the authors and in the future for the readers, it will be very important to improve this introductive section of this manuscript, by providing a short premise about biomaterials employed in regenerative procedures with different and significant clinical outcomes.

LINE 69: The PRP is obtained from the heparinization, which makes it incoagulable of the blood taken from the patient and can be stored for a few days at −19 °C, or it can be prepared immediately after the blood collection. In any case, before centrifuging the blood (the protocol includes two centrifugations) to remove the red blood cells, anticoagulant factors such as bovine thrombin or CaCl2 must be added to trigger the coagulation and fibrin formation cascade. This process leads to a tumultuous fibrin network formation process, resulting in somewhat irregular fibrin. PRP should also be used within 4 h of isolation, as growth factors are secreted within 10 min of preparation and reach 95% after 1 h. This type of platelet concentrate presents disadvantages: firstly, the presence of anticoagulants such as bovine thrombin could cause allergic reactions and coagulopathies due to the action of antibodies against factors V, XI, and consequent thrombus formation. Furthermore, the final preparation without rigidity requires the further addition of bone grafts to maintain a stable volume (see, for reference: https://doi.org/10.3390/biomedicines10020218). This is the major concern of this manuscript: according also to recent scientific evidence, the authors should provide detailed description about the disadvantages associated to PRP.

Specifically, this is the sentence, which I refer to:

“The PRP induced tissue repair, regeneration, and wound healing potential is based on the release of a plethora of platelet constituents from alpha, dense, and lysosomal granules, as well as the activities of many platelet adhesion molecules”.

Anyway, I highlighted in yellow this sentence, because before it in this part of the manuscript it will be useful to provide a very brief description about the procedures that lead to the development of PRP, according to recent evidence on this topic. In order to preserve the original plan and structure of this section of the manuscript, it is not necessary to refer in a deep way to bovine thrombin and bone grafts. For these reasons, and as previously suggested, the priority of this section is to provide a complete and short description of the procedures, which underlying PRP development. The authors should only limit themselves to describing the procedures underlying to PRP development, always according to recent scientific evidence.

The potential of platelet-rich plasma (PRP), a kind of blood-derived product that is produced via centrifugation or the apheresis process for the platelet enrichment of plasma from autologous or allogenic blood has attracted substantial interest in regenerative medicine. In fact, some recent studies suggested also the ability of PRP extracellular vesicles to transfer microRNA, like (miR)-130a-3p or miR-217, with the aim to alleviate myocardial ischemia and regulate inflammation (see, for reference: https://doi.org/10.1016/j.cellsig.2024.111106). It will be interesting to mention also the linkage between PRP and microRNA, in the regulation of different inflammatory pathways. This is for me a very minor concern of this manuscript and probably, it will be useful only to mention this association, as a future interesting perspective of the research. This was the main aim of my suggestion: this is just a preliminary, and a future step of the research, and there won’t be problems, if the author will prefer to not discuss in depth.  

The main topic is interesting, and certainly of great clinical impact. This is a significant contribute to the ongoing research on this topic, as it extends the research field on the regulatory role of a multipurpose biological platform comprised of high-definition PRP and viable protein-rich plasma, as well as their typical biological contributions in tissue repair, regeneration, and wound healing. Furthermore, the authors also give their deep insight for some works.

As regards the scientific background of this manuscript, this manuscript relies on a multitude of morphological and molecular analysis, to derive its conclusions.

The conclusion of this manuscript is perfectly in line with the main purpose of the paper: the authors have designed and conducted the study properly. As regards the conclusions, they are well written and present an adequate balance between the description of previous findings and the results presented by the authors.

Finally, this manuscript also shows a basic structure, properly divided and looks like very informative on this topic. Furthermore, figures and tables are complete, organized in an organic manner and easy to read.

In conclusion, this manuscript is densely presented and well organized, based on well-synthetized evidence. The authors were lucid in their style of writing, making it easy to read and understand the message, portrayed in the manuscript. Besides, the methodology design was appropriately implemented within the study. However, many of the topics are very concisely covered. This manuscript provided a comprehensive analysis of current knowledge in this field. Moreover, this research has futuristic importance and could be potential for future research. However, major concerns of this manuscript are with the introductive section: for these reasons, I have major comments for this section, for improvement before acceptance for publication. The article is accurate and provides relevant information on the topic and I have some major points to make, that may help to improve the quality of the current manuscript and maximize its scientific impact. I would accept this manuscript if the comments are addressed properly.

Author Response

Dear Reviewer 3.

Thanks for your time to review our manuscript. The authors appreciate your work and knowledge on the subject.

In general, we decided to modify the title to Protein Rich Platelet Rich Plasma (PR-PRP) instead of protein rich platelet concentrate, as we the authors believe this is a more representative and accurate definition according to the described PRP formulations described in this manuscript. PRPC has been changed in entire manuscript to PR-PRP. We hope you agree.

Based on your review we address here some of your comments separately.

Line 59 comment:

We added the following paragraph based on your comments and added several references:

“Over the last several decades, biologically derived materials and tissues have evolved in many different forms, like coatings, scaffolds, biosensors, and functional biomaterials (1,2). Initially, it was thought to substitute damaged human body parts. Later, the focus shifted from the use of materials to biology, giving rise to the development of tissue engineering concepts (3–5).

Regenerative medicine technology and clinical applications emerged from several clinical practices like bone grafting, surgical implants, scaffold-based biomaterials, and bone marrow transplantations (1). The use of autologous biological preparations, including PRP and fibrin-based scaffolds, have gained remarkable momentum”

Line 69 comment:

We added the following paragraph based on your comments and added several references:

The term “platelet‐rich plasma” was first used in 1954 by Kingsley et al. and referred to the standardization of platelet concentrate preparations for transfusion (9).  In 1972, for the first time, Matras used platelets as sealants to improve tissue healing following surgical procedures (10). Thereafter, an autologous product termed “platelet–fibrinogen–thrombin mixture” was developed, including, in fibrin glue, a significant concentration of platelets, to reinforce fibrin polymerization in corneal wounds (11). In the following years, Knighton et al. described the role of platelets in wound healing procedures by using a blood-derived product called “platelet-derived wound healing factors” (12). The term platelet gel was introduced by Whitman and associates (13) in oral and maxillofacial surgery while the term PRP in regenerative medicine was introduced by Marx et al. in 1998, identifying the potential for tissue healing using platelet growth factors (14). Thereafter, the term PRP was generically associated many different formulations of platelet concentrates, including the denser fibrin based PRP (15).

Comment on miRNA:

As highlighted in my previous report, there are some recent studies, which analyzed the association between PRP and microRNA. This is for me a very minor concern of this manuscript and probably, it will be useful only to mention this association, as a future interesting perspective of the research. This was the main aim of my suggestion: this is just a preliminary, and a future step of the research, and there won’t be problems, if the author will prefer to not discuss in depth.

The authors agree that associations between PRP and miRNA have been published but we feel that adding comments regarding this topic would be overwhelming in this extensive manuscript. We hope you agree with us.

Thank you very much for your thorough review and we are convinced that your comments contributed to a more complete manuscript

Thank you for your time of reviewing, on behalf of the authors.

Round 2

Reviewer 3 Report

Comments and Suggestions for Authors

The authors have improved the scientific impact and quality of this manuscript.